# Infrared Spectroscopic Study of Multi-Component Lipid Systems: A Closer Approximation to Biological Membrane Fluidity

**DOI:** 10.3390/membranes12050534

**Published:** 2022-05-20

**Authors:** Maria C. Klaiss-Luna, Marcela Manrique-Moreno

**Affiliations:** Chemistry Institute, Faculty of Exact and Natural Sciences, University of Antioquia, A.A. 1226, Medellin 050010, Colombia; maria.klaiss@udea.edu.co

**Keywords:** lipid model membranes, supported lipid bilayers, LL-37, thermal behavior, membrane fluidity, infrared spectroscopy

## Abstract

Membranes are essential to cellular organisms, and play several roles in cellular protection as well as in the control and transport of nutrients. One of the most critical membrane properties is fluidity, which has been extensively studied, using mainly single component systems. In this study, we used Fourier transform infrared spectroscopy to evaluate the thermal behavior of multi-component supported lipid bilayers that mimic the membrane composition of tumoral and non-tumoral cell membranes, as well as microorganisms such as *Escherichia coli*, *Pseudomonas aeruginosa*, *Staphylococcus aureus*. The results showed that, for tumoral and non-tumoral membrane models, the presence of cholesterol induced a loss of cooperativity of the transition. However, in the absence of cholesterol, the transitions of the multi-component lipid systems had sigmoidal curves where the gel and fluid phases are evident and where main transition temperatures were possible to determine. Additionally, the possibility of designing multi-component lipid systems showed the potential to obtain several microorganism models, including changes in the cardiolipin content associated with the resistance mechanism in *Staphylococcus aureus*. Finally, the potential use of multi-component lipid systems in the determination of the conformational change of the antimicrobial peptide LL-37 was studied. The results showed that LL-37 underwent a conformational change when interacting with *Staphylococcus aureus* models, instead of with the erythrocyte membrane model. The results showed the versatile applications of multi-component lipid systems studied by Fourier transform infrared spectroscopy.

## 1. Introduction

All cellular organisms have a thin, fluid, and flexible membrane based on lipids that define them as units, separates them from their surroundings, and participates in several physiological processes [1]. Biological membranes have diverse functions and compositions; however, they have a common and critical property called fluidity [2,3,4]. Membrane fluidity is a complex concept that from a physicochemical view is an elastomechanical property. However, it includes several parameters such as structure, composition and disposition of membrane lipids that contribute to the state of the membrane. The complex diversity of lipid types creates a delicate balance that regulates the structural properties of biomembranes, and therefore the environment for proteins and other membrane components, and this is associated with the flexibility needed for cellular morphological transformation including division, differentiation, and general adaptation for each kind of membrane [5,6]. These processes are associated with molecular motion within a lipid bilayer and activity regulation of other molecules, including membrane-bound proteins. It was reported that abnormalities in the physical properties of cell membranes may underlie the defects that are strongly linked to hypertension, stroke and other cardiovascular diseases [7,8], diabetes mellitus [9], and Alzheimers [10]. Therefore, studies of the relationships between fluidity and phase transitions in membranes are relevant. For this reason, there has been a great interest in the use of artificial model lipid membranes with a simplified composition to study membrane properties, structure, and activity of natural or synthetic compounds [11].

There are several techniques used for studying a wide variety of temperature-induced transitions in biological systems such as Differential Scanning Calorimetry (DSC) [12]. DSC is a non-perturbing thermodynamic technique that measures the heat exchange associated with cooperative lipid phase transitions in model and biological membranes. This technique is useful not only in determining thermotropic phase behavior (T_m_) and energy, but also in obtaining information on the cooperativity of the transitions [13]. The most important parameters obtained by DSC are the change in the main phase transition temperature, the width of the main transition peak (cooperativity) and the enthalpy change (ΔH) associated with the process. One of the most common techniques used for studying a membrane state is fluorescence spectroscopy. The technique is based on the use of membrane probes such as 6-dodecanoyl-2-dimethylaminonaphthalene (laurdan) and 1,6-diphenyl-1,3,5-hexatriene (DPH) [14,15]. The Laurdan fluorescent spectrum is sensitive to polarity (hydration level), dipolar dynamics of the environment and the phase state of phospholipid bilayers [16]. Therefore, laurdan provides information about molecular dynamics at the level of the glycerol backbone and can distinguish whether a membrane is in a gel or liquid-crystalline state. The laurdan molecule is strongly anchored in the hydrophobic core of the bilayer by hydrophobic interactions between its lauric acid tail and the lipid alkyl tails while the fluorescent moiety is located at the glycerol level of the phospholipid headgroups [17]. The fluorescent probe DPH was widely used to study the structure and dynamic properties of the model and natural membranes. DPH was originally proposed as a probe to estimate the microviscosity of cell membranes and the rigidity of binding sites on proteins [14].

Another technique extensively used in the study of membrane properties is Fourier transform infrared spectroscopy (FT-IR). Chemical bonds undergo different forms of vibrations such as stretching, twisting and rotating. The energy of most molecular vibrations corresponds to the infrared region of the electromagnetic spectrum. Infrared spectroscopy measures these vibrations and provides information about molecular structure and structural interactions. One of the most important advantages of FT-IR spectroscopy is that experiments with biomolecules and without the addition of probes can be performed in diverse environments such as water, thin films, organic solvents, detergent micelles and lipid bilayer matrix [18]. Applications such as attenuated total reflection (ATR) provide rapid and sensitive monitoring of one of the most important physicochemical parameters of membranes, fluidity. If an exogenous molecule interacts with the phospholipid bilayer, changes can be detected and monitored [18].

However, the simplification of model membranes used in these techniques was extensively questioned on the basis that it does not represent the membrane properly, especially the fluidity, charge, and lipid headgroup complexity [19]. For this reason, the aim of this study was to use multi-component lipid systems in order to understand the direct relationship of membrane lipid composition to fluidity, and how these complex systems represent a better approximation between Gram-negative and Gram-positive bacteria and, in eukaryotes, between cancer and non-cancer cell membranes. Additionally, we used two multi-component lipid systems that represent the bacterial and erythrocyte cell membranes to study the conformational change undergone by a recognized peptide such as LL-37. Human cathelicidin LL-37 is a well-studied peptide with a + 6 charge at pH 7.4 that has several physiological roles in the body, being recognized for its antimicrobial, antifungal and antiviral activities [20,21,22,23]. When the peptide interacts with membranes it assumes an α-helical structure that was reported by circular dichroism [24,25].

## 2. Materials and Methods

### 2.1. Chemical Reagents

1,2-dimyristoyl-sn-glycero-3-phosphoglycerol sodium salt (DMPG, Lot. 140PG-167), 1’,3’-bis [1,2-dimyristoleoyl-sn-glycero-3-phospho]-glycerol sodium salt (CL, Lot. 750332P-200MG-A-030), 1,2-dimyristoyl-sn-glycero-3-phosphoethanolamine (DMPE, Lot. 140PE-63), 1,2-dipalmitoyl-sn-glycero-3-phosphocholine (DPPC, Lot. 160PC-318), Sphingomyelin Egg Chicken (SM, Lot. 860061P-25MG-A-116), 1,2-dipalmitoyl-sn-glycero-3-phosphoethanolamine (DPPE, Lot. 160PE-106), 1,2-dipalmitoyl-sn-glycero-3-phospho-L-serine (sodium salt), (DPPS, Lot. 840037P-500MG-A-078CL750332P-200MG-A-030) and Cholesterol (CH, Lot. CH-92) were purchased from Avanti Polar Lipids (Alabaster, AL, USA). Breast cancer cell line MCF-7 (HTB-22™) was purchased from ATCC^®^ (Manassas, VA, USA). HPLC-grade methanol and chloroform were purchased from Merck (Kenilworth, NJ, USA). HEPES was purchased from Sigma-Aldrich (St. Louis, MO, USA), NaCl from Carlo Erba (Val de Reuil, NOR, FR) and EDTA from Amresco (Solon, OH, USA).

LL-37 peptide (LLGDFFRKSKEKIGKEFKRIVQRIKDFLRNLVPRTES, Lot. V1440EE070/PE1324) was purchased from GenScript (Piscataway Township, NJ, USA) and synthesized according to the sequence by solid-phase method. The purity of the peptide was determined to be higher than 95% by analytical HPLC, TFA removal was performed, and the molecular weight was confirmed with MALDI-TOF mass spectrometry.

### 2.2. Cell Cultures and Lipid Extraction

MCF-7 breast cancer cell line (ATCC HTB-22^™^) was cultured in Dulbecco’s modified Eagle’s medium (DMEM), supplemented with 5% fetal calf serum, 100 µg/mL penicillin and 100 µg/mL streptomycin. The cells were grown at 37 °C in a humidified incubator with 5% CO_2_/95% air. Cell cultures were examined under a microscope for correct morphology, adherence, and exponential growth. To extract the lipids, cells were trypsinized, pelleted, washed with 2 mL of water and re-centrifuged at 6000 rpm for 15 min at 4 °C in 15 mL polyethylene centrifuge tubes. The supernatant was discarded, while the pellet was freeze-dried (Chamber temperature −60 °C, SP Scientific, Gardiner, NY, USA, pressure < 7 × 10^−1^ mbar).

Lipid extraction was performed according to the two-step Bligh and Dyer lipid extraction method suitable for samples in incubation medium, tissue or cell suspensions [26]. A total of 65 mg of MCF-7 cells were weighed in a clean test tube and resuspended in 5 mL deionized water, and the mixture was then vortexed for homogenization. The cell suspension was quantitatively transferred into a clean glass separatory funnel. The tube was rinsed with 5 mL deionized water. A total volume of 96 mL of solvent consisting of chloroform, methanol and water (1:2:0.8; *v*/*v*/*v*) was then added. The samples were shaken for 20 s immediately after the solvent had been added and allowed to stand for about 18 h, with occasional shaking. Phase separation of the biomass–solvent mixtures in the separatory funnels involved adding chloroform and water to each separation funnel to obtain a final chloroform–methanol–water ratio of 1:1:0.4 (*v*/*v*/*v*). The chloroform phase was transferred into a clean 50 mL flask bottle to remove the solvent and concentrate the samples, which were then washed 3 times at the same volume of 0.9% NaCl (*w*/*v*). The organic phase containing the lipid extract was collected and concentrated by evaporating the solvent under a stream of nitrogen.

### 2.3. Phase Transition Measurements by Infrared Spectroscopy

Supported lipid bilayers (SLBs) were prepared in situ in a BioATR II cell. The unit was integrated with a Tensor II spectrometer (Bruker Optics, Ettlingen, Germany) with a liquid nitrogen MCT detector using a spectral resolution of 4 cm^−1^ and 120 scans per spectrum. The desired temperature was set by a Huber Ministat 125 computer-controlled circulating water bath (Huber, Offenburg, Germany) with an accuracy of ±0.1 °C. First, the background was taken using 20 mM HEPES buffer, 500 mM NaCl and 1 mM EDTA in the same temperature range. Subsequently, to coat the silicon crystal, stock solutions of the different lipid systems were dissolved in chloroform. The preparation of stock solutions was carried out depending on the lipid system to analyze. For non-tumoral and tumoral membrane composition this was DPPC/SM/DPPE 4.35:4.35:1 (*w*/*w*) and DPPC/SM/DPPE/DPPS/3.85:3.85:0.8:1.5 (*w*/*w*), respectively [27,28]. For *Escherichia coli* (*E. coli*) lipid system it was PE:PG:CL 75:20:5 [29]. For *Pseudomonas aeruginosa (P. aeruginosa)* lipid system it was PE:PG:CL 65:23:12 (*w*/*w*) [30]. Finally, for *Staphylococcus aureus (S. aureus)* lipid system it was DMPG:CL 80:20 (*w*/*w*) [31].

The cell was filled with 20 µL of the lipid stock solution, and the chloroform was evaporated, resulting in a lipid multilayer film. For in situ measurements the cell was subsequently filled with 20 µL of buffer or peptide solution and incubated over the phase transition temperature for 10 min. To determine the position of the vibrational band in the range of the second derivative of the spectra, all the absorbance spectra were cut in the 2970–2820 cm^−1^ range, shifted to a zero baseline and the peak picking function included in OPUS software. The results were plotted as a function of the temperature. To determine the transition temperature (T_m_) of the lipids, the curve was fitted according to the Boltzmann model to calculate the inflection point of the obtained thermal transition curves using the OriginPro 8.0 software (OriginLab Corporation, Northampton, MA, USA).

### 2.4. Determination of the Secondary Structure of LL-37 Using Multi-Component Lipid Systems

LL-37 Peptide solution was prepared at 1mM concentration in buffer (10 mM HEPES, 500 mM NaCl, 1 mM EDTA, pH 7.4). Appropriate amounts of DMPC:SM:DMPE (22:20:6) for erythrocyte membrane [32], and DMPG:CL (80:20) for *S. aureus* membrane [31] were weighed in order to obtain a 5 mM final concentration of representative liposomes. Lipids were dissolved in pure chloroform in a glass test tube, the solvent was dried under a stream of nitrogen and the traces were removed by keeping the samples under reduced pressure (about 13.3 Pa) for 30 min. Dried lipids were hydrated in buffer. Multilamellar vesicles (MLVs) were formed by sonicating the samples above the main phase transition temperature of the lipids for at least 15 min. For the determination of the secondary structure, LL-37 was added to the liposome suspension to obtain a 15 molar% concentration. The experiments were performed at 37 °C in an AquaSpec Cell (Bruker Optics, Ettlingen, Germany). The unit was integrated with a Tensor II spectrometer with a liquid nitrogen MCT using a spectral resolution of 4 cm^−1^ and 120 scans per spectrum. The secondary structure elements α-helix and β-sheet were predicted following the methods supplied by the Confocheck^TM^ system (Bruker Optics, Ettlingen, Germany). These methods calculate the secondary structure with a multivariate partial-least-squares algorithm (PLS) based on a calibration data set of 45 different proteins.

## 3. Results

### 3.1. Phase Transition Experiments by Infrared Spectroscopy

One of the most important physicochemical parameters to follow the lipid order and packing of the hydrophobic core of membranes is the wavenumber peak position of the ν_s_CH_2_ band. Depending on the temperature, the lipid bilayer has different states. At lower temperatures, in the gel phase, ν_s_CH_2_ lies around 2850 cm^−1^; while at higher temperatures, in the liquid crystalline phase, the band lies around 2853 cm^−1^. Following the phase transition, it is possible to calculate the main transition temperature (T_m_) of a lipid system. The results of the phase transition measurements of four pure lipid systems are summarized in Figure 1. The change in the peak position of the symmetric stretching vibration band of the methylene group as a function of the temperature of the four pure lipids produced a sigmoidal curve, which reflects the high degree of molecular aggregation of these systems. The transitions of saturated phospholipids were shown to be highly cooperative events, with transition ranges of less than 0.1 °C. Afterward, multi-component lipid systems to mimic two cell membrane systems were prepared, considering the reported composition for cancer PC:SM:PE:PS (3.85:3.85:0.8:1.5 *w*/*w*), and non-cancer membranes PC:SM:PE (4.35:4.35:1 *w*/*w*) proposed by Li et al. [33] and Yeung and collaborators [28]. The results obtained by FT-IR are summarized in Figure 2. As expected, the phase transition for the multi-component lipid systems is broader than for the pure systems. However, obtaining a sigmoidal curve allows the same method as for pure lipid systems to be applied in order to determine the phase transition temperature through Boltzmann fitting. The T_m_ obtained for non-cancer and cancer cell membranes are 39.9 and 41.0 °C, respectively.

One important component of the eukaryotic cell membrane is cholesterol. For this reason, we evaluated a multi-component lipid system including cholesterol. Figure 3 shows the thermal behavior of the non-cancer model including cholesterol and the comparison of this with the thermal behavior of the total lipid extract from breast cancer cell line MCF-7. When cholesterol was included in the multi-component lipid system, the characteristic sigmoidal curve was lost. Analysis of the results of the phase transitions of multi-component lipid system and MCF-7 lipid extract showed that both systems exhibited similar thermal behavior, and the transitions obtained were more lineal than sigmoidal.

The results showed that transitions are broader than the previous results obtained in Figure 2. Nonetheless, there is a remarkable difference in the peak positions of the ν_s_CH_2_ vibration; the vibration of the lipid extract is located at higher frequencies than the ν_s_CH_2_ vibrations of the synthetic model, showing that the lipid extract is more fluid.

As part of our results, we also evaluated the thermal behavior of representative multi-component lipid systems of two Gram-negative bacterial membranes *P. aeruginosa* [30], and *E. coli* [29]. The results of these two lipid systems are represented in Figure 4. The results showed a very similar T_m_ of 45.6 and 45.9 °C for *P. aeruginosa* and *E. coli*, respectively. However, the SLBs behave differently in the two models, considering that the *P. aeruginosa* and *E. coli* systems have different proportions of PG:CL, lipids related with the electrostatic properties and the negative charge of the membrane surface in each model.

Finally, we studied a multi-component lipid system representative of *S. aureus*. The results of this are summarized in Figure 5. The pure lipid systems of PG and CL are included in the graph to compare the results obtained with the different proportions of PG:CL evaluated. The first model was the sensitive *S. aureus*, based on our previous results on lipid quantification of total lipids of *S. aureus* strain RN4220. We reported a ratio of PG:CL (80:20) [31]. However, for antibiotic-resistant and other *S. aureus* strains, different PG:CL ratios, such as 70:30 and 60:40, were reported [34,35]. As can be observed in the results, the extremes in the graph correspond to the pure PG and CL lipids. From the left side of the graph, increasing concentrations of CL caused an increase in the T_m_ of the model lipid system from 28.8 °C in the sensitive *S. aureus* model to 38.3 °C in the resistant model. Additionally, the increasing concentrations of CL in the gel phase for all the lipid systems induced a reduction in the ν_s_CH_2_ vibration to lower wavenumbers.

### 3.2. Determination of the Secondary Structure of LL-37

Given that the structural conformation of antimicrobial peptides changes through interaction with membranes, we evaluated the conformational change in the human cathelicidin LL-37, a very highly-studied peptide, in two representative lipid systems. The obtained results are summarized in Table 1. Analysis of the secondary structure found that in the buffer, the LL-37 peptide was partially helical. In the case of the representative multi-component lipid system of erythrocyte membrane (DMPC:SM:DMPE) the results showed a conformational change of 8% of the LL-37 helical structure. However, when the peptide interacted with the bacterial model of *S. aureus* (DMPG:CL) the peptide acquired 63.1% of its secondary structure as an α-helix.

## 4. Discussion

Membranes are fluid, heterogeneous and dynamic systems. Fluidity is related to the viscosity of the lipid membrane, which is an important mechanical property of the cell membrane. It is an intensive and bulk property related to the translational, rotational, and vibrational movement of membrane lipids, with important consequences for other molecules inserted in the hydrophobic core such as proteins [36,37]. Increased fluidity enhances the free movement of phospholipid molecules and protein moieties in the membrane to facilitate various biological functions including ion transport, cell signaling and cell growth [10]. Fluidity is affected by lipid chemical structure, temperature, and cholesterol content [38]. In the case of fatty acids, fluidity has a characteristic value depending on the unsaturation and length of the acyl chains [39], and on the structure of the head groups of the phospholipid [40]. At low temperatures, the hydrocarbon chains of lipids are organized in an orderly arrangement called the gel state. In contrast, with increasing temperature, lipid molecules vibrate faster, causing fusion that leads to a liquid crystalline state, which is more fluid and disordered [2]. These phases consist of hydrated phospholipid aggregates. The aggregation process is driven by the hydrophobic effect of acyl chains. Therefore, transitions between phases can be induced by varying temperatures. For pure lipid systems, the temperature value where the lipids change state is called the main transition temperature (T_m_). For multi-component lipid mixtures, T_m_ will correspond to a value depending on the participating lipids and their proportions. This thermal process is highly cooperative and easy to monitor using several techniques. Cholesterol has been extensively studied given its considerable effects on the conformation, fluidity and thermotropic properties of membranes [41,42,43,44]. Additionally, malignant cells have been described as more fluid, due to their lower cholesterol content than normal cells. This lower cholesterol content makes malignant cells more susceptible to membrane acting-drugs by facilitating the destabilization of the membrane [45].

One of the most versatile techniques to follow thermal behavior is Fourier transformed infrared spectroscopy. This allows monitoring of the order of the lipid acyl chains in terms of the symmetric vibration of CH_2_ of phospholipid molecules and the changes due to temperature, which is a measurement of fluidity. Low frequencies of methylene groups are associated with a less mobile phase and high frequencies with a high mobile state. In addition, this technique can be used to study the effects of changes in pH, solute concentration, and the interaction with exogenous agents [46] in the absence and presence of an agent as a function of temperature and at different lipid:compound ratios. Since biological membranes are very complex systems given the current understanding of lipid behavior, model membranes have been extensively used for study purposes instead of natural membranes. The most well-known biomimetic systems for such purposes are lipid monolayers, lipid vesicles, and supported lipid bilayers [47]. These models have been extensively studied using different biophysical techniques. These analytical or spectroscopic techniques are based on the fact that molecules can produce changes in the physical and thermodynamic properties of model membranes [48].

We worked with SLBs as model membranes to study the thermal behavior of several lipid systems. Analysis was carried out on symmetric stretching CH_2_ vibration through FT-IR as a function of temperature. The first step in the evaluation was obtaining the phase transitions of the pure lipids DPPC, DPPE, DPPS, and SM. The results of the pure lipid systems were sigmoidal functions representing the transitions from gel to the liquid-crystalline states. The temperature-dependent changes in the ν_s_CH_2_ stretching band allowed the determination of the phase transition temperature for the studied SLBs. This method was based on calculating the second derivative of the sigmoidal curves. The results of the determination of the T_m_ were consistent with data previously reported in the literature using differential scanning calorimetry (DSC) [49,50,51,52], which is a very precise calorimetric technique to obtain the phase transitions of lipid suspensions [53].

Although pure lipid systems such as PC and PS have been extensively used to represent non-cancer and cancer cell membranes, respectively [54,55,56], these simplified models are far from representing the composition of a complex biological membrane. For this reason, the next step was to prepare multi-component lipid systems representative of both membranes. We used the composition reported by Almarwani et al. for the normal cell membrane, and an anionic phospholipid range of 10–20% for the cancer model, without including cholesterol in the multi-component lipid model [28]. In the absence of cholesterol, slightly broader sigmoidal curves were obtained, through which it was possible to calculate the T_m_ of the systems. The results showed that the temperatures obtained for non-tumoral and tumoral cells were 39.9 and 41.0 °C, respectively. These results could be considered similar, but the charge of the systems is not. The cancer model is slightly but not completely negatively charged at the surface, in comparison to a single-component system built exclusively from PS. The phosphatidylserine in the multi-component system is in principle distributed along the surface of the model membrane, which is more representative of a cancer cell membrane. It was extensively reported that malignant cells of several types of cancer lose their asymmetric distribution [57,58], which results in the exposure of the negatively charged PS on the surface of their membranes. In the case of the non-cancer cell membrane, there is no charge at the membrane surface since all lipids used in the model were zwitterionic. This is also in accordance with the typical non-charge of a eukaryotic membrane but has the headgroup complexity from it. Our results represented the different fluidity of cancer cell membranes in comparison with non-cancer cell membranes. This characteristic can be monitored through the analysis of the wavenumber at a specific temperature, for example at 37 °C, which is 2850.8 for the cancer model membrane, and 2850.5 cm^−1^ for the non-cancer membrane. Our findings are in accordance with the previously-described characteristic of cancer cell membranes as more fluid systems than normal cell membranes [4,59].

Our attempt to include cholesterol in the multi-component lipid systems resulted in a very broad transition, where the inflection point characteristic of the gel to liquid-crystalline change was not evident. This transition was compared with the one obtained from the total lipid extract from MCF-7 cells. The result was very similar, consisting of a very broad phase transition. The main difference was that the ν_s_CH_2_ stretching vibrations from the lipid extract were located at higher frequencies, which could suggest that the extract contains unsaturated lipids and higher cholesterol content. The broadening effect of the cholesterol in the phase transition is not surprising, as cholesterol acts as a phase modulator in mammalian membranes, generating modifications in membrane dynamics and fluidity [41,51].

Multi-component lipid systems representative of *P. aeruginosa* (PE:PG:CL; 65:23:12 *w*/*w*) and *E. coli* (PE:PG:CL; 75:20:5 *w*/*w*) were also evaluated to identify the T_m_ of the systems. Both are Gram-negative bacteria and represent a major health problem worldwide. They are related to several infections and are difficult to treat due to the absence of new antimicrobial agents active against this group of pathogens [60,61]. The results obtained showed very similar values for T_m_, which is closely related to the similar composition of the cell membranes. However, the technique is sensitive enough to detect small variations in the lipid composition of the systems, resulting in slight differences in the T_m_ obtained of 45.6 and 45.9 °C for *P. aeruginosa* and *E. coli*, respectively.

Another very interesting system to study was *S. aureus* model membrane. In this Gram-positive bacteria, two lipids were identified as major components, phosphatidylglycerol (PG) and cardiolipin (CL) [31]. However, the role of CL is not completely understood. Several authors have suggested that increasing concentrations of CL in *S. aureus* membrane is associated with a resistance mechanism based on the modification of membrane composition in the bacteria [62,63]. The polar headgroups of the cardiolipin structure have a relatively small size, which promotes greater cohesion between the CL hydrocarbon chains [64]. For this reason, it was proposed that higher concentrations of the lipid in the bilayer increase the membrane packing due to an increase in the lateral density of fatty acids. Some studies have suggested that an elevated CL content in the membrane of *S. aureus* may contribute to bacterial resistance to antibiotics, including Daptomycin [65]. Considering the relevant physical–chemical properties of CL in *S. aureus* membrane, we evaluated different proportions of CL in the SLBs by FT-IR. The results suggest that CL plays an important role in the membrane characteristics of *S. aureus*. Specifically, they showed that increasing concentrations of CL have an effect on the fluidity of the system, increasing the T_m_ of the model membranes and a fixed temperature. For example, the ν_s_CH_2_ vibration of the system 60:40 appeared at wavenumbers lower than 70:30 or 80:20. These results provide evidence that CL imposes structural consequences on the bilayer core of the *S. aureus* membrane.

Finally, a very attractive application of the multi-component lipid systems is the determination of the secondary structure of bioactive peptides. The conformational change that peptides undergo when they interact with the cell membrane has been extensively studied [66,67]. The most extensively used technique for the determination of the secondary structure is circular dichroism. In this technique solvents such as trifluoroethanol are used to represent the membrane environment. However, the possibility of using multi-component lipids systems that can be designed depending on the membrane under study, opens in turn the possibility of using more representative model systems to study the conformational change. Most peptides change conformation from random coiled to an α-helical structure. In this study the very highly-studied peptide LL-37 was made to interact with two different multi-component lipid systems, representative of erythrocytes and *S. aureus* membranes, to evaluate the conformational change depending on the composition of the model membrane used. It was demonstrated that human cathelicidin LL-37 after binding to the membrane assumes an α-helical structure [24]. This peptide has several physiological roles in the body, being recognized for its antimicrobial, antifungal and antiviral activities [20,21,22,68,69,70]. The results showed that, in solution, LL-37 was partially helical. This result could be explained by the fact that cathelicidins often show a slightly α-helical structure at low concentrations in buffer solution [25]. In the presence of liposomes representative of the erythrocyte membrane the peptide underwent a mild conformational change, but in the presence of the liposomes representative of *S. aureus* membrane the change in conformation was very strong. LL-37 peptide is known to interact with negatively charged phospholipid vesicles, leading to the induction of a secondary structure [24,71], which is highly associated with its biological activity.

## 5. Conclusions

The complexity of the cell membrane makes it a very intricate system to study in its natural state. However, given the importance of understanding how molecules interact and affect the physicochemical properties of membranes, it is necessary to use representative models of these systems. Our results demonstrated the possibility of using multi-component lipid systems to monitor membrane fluidity by infrared spectroscopy. This application opens the possibility of studying the thermal behavior of eukaryotic and bacterial model membranes, where the lipid composition and charge mainly influence the phase transition temperature. The results of this study have led to the identification of differences in both tumoral and non-tumoral membrane fluidity, and even to comparisons between the thermal behavior of synthetic and natural lipids, thereby allowing the main effect of cholesterol on the phase transition to be appreciated. The results also showed that, despite the similarity in the phospholipid composition between Gram-positive and Gram-negative bacteria, proportions established significant differences in the lipid system charge, thereby conditioning the T_m_ parameter. We finally wanted to highlight one more application of multi-component lipid systems for the conformational change that peptides can undergo in the presence of a lipid environment. Thus, the helical content of the recognized LL-37 peptide was evaluated in aqueous, erythrocyte, and bacterial model membranes. The folding trend is consistent with reports using circular dichroism. As a future perspective, further investigations with multi-component lipid systems will be used as a strategy to study the effect of potential antimicrobial and antitumoral agents in the biophysical properties of eukaryotic and bacterial membranes, as well as the secondary structure prediction of peptides in those lipid environments.

## Figures and Tables

**Figure 1 membranes-12-00534-f001:**
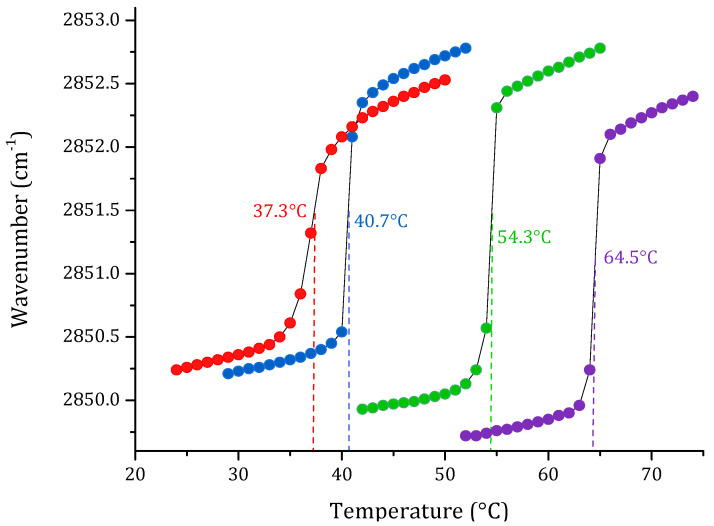
Peak positions of the ν_s_CH_2_ vibration bands of the methylene groups as a function of the temperature of the pure supported lipid bilayers of DPPC (●), DPPS (●), DPPE (●) and SM (●) in buffer (10 mM HEPES, 500 mM NaCl, 1 mM EDTA, pH 7.4).

**Figure 2 membranes-12-00534-f002:**
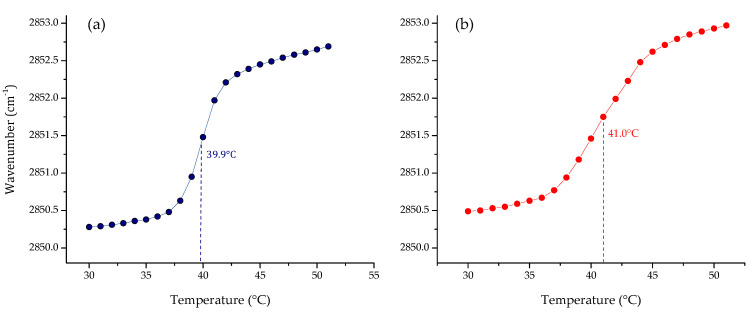
Peak positions of the ν_s_CH_2_ vibration bands of the methylene groups as a function of temperature of the representative model of (**a**) non-cancer (PC:SM:PE; 4.35:4.35:1 *w*/*w*) and (**b**) cancer cell membranes (PC:SM:PE:PS; 3.85:3.85:0.8:1.5 *w*/*w*) in buffer (10 mM HEPES, 500 mM NaCl, 1 mM EDTA, pH 7.4).

**Figure 3 membranes-12-00534-f003:**
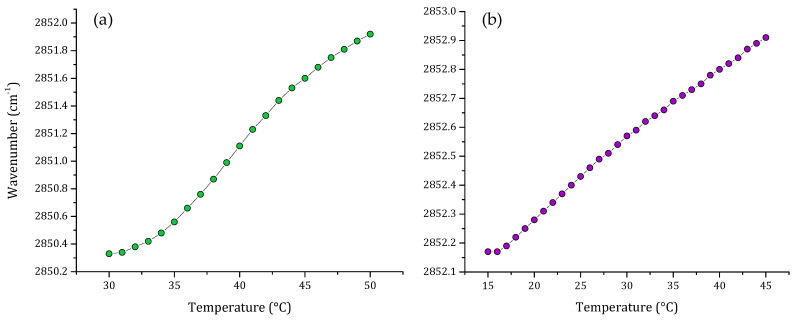
Peak positions of the ν_s_CH_2_ vibration bands of the methylene groups as a function of temperature of (**a**) non-cancer cell model membrane (PC:SM:PE:CH; 4.35:4.35:1:1 *w*/*w*) and (**b**) lipid extract of breast cancer MCF-7 cell line, both transitions were obtained using buffer (10 mM HEPES, 500 mM NaCl, 1 mM EDTA, pH 7.4).

**Figure 4 membranes-12-00534-f004:**
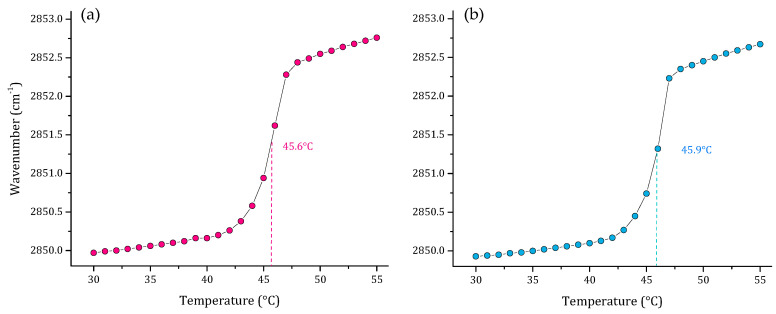
Peak positions of the ν_s_CH_2_ vibration bands of the methylene groups as a function of temperature of (**a**) *P. aeruginosa* PE:PG:CL 65:23:12 (*w*/*w*) and (**b**) *E. coli* PE:PG:CL 75:20:5 (*w*/*w*) multi-component lipid systems in 10 mM HEPES, 500 mM NaCl, 1 mM EDTA, pH 7.4.

**Figure 5 membranes-12-00534-f005:**
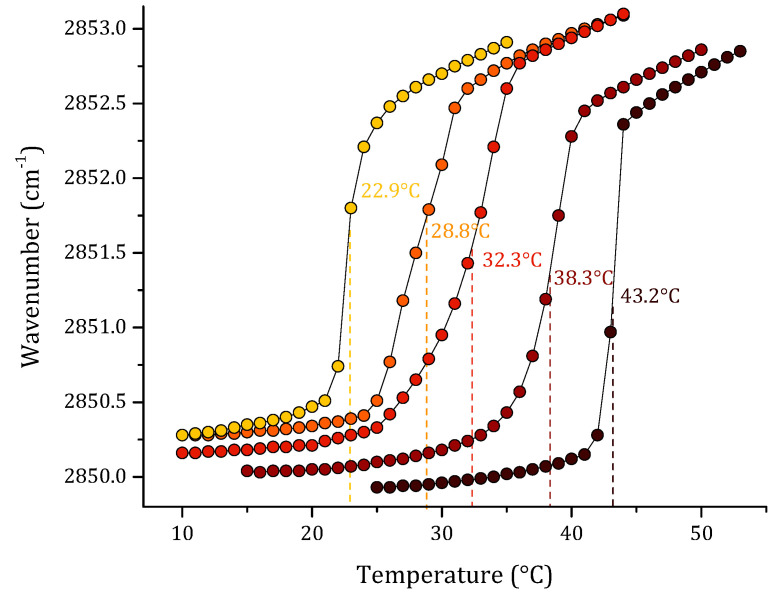
Peak positions of the ν_s_CH_2_ vibration bands of the methylene groups as a function of temperature of (●) DMPG, (●) DMPG:CL (80:20), (●) DMPG:CL (70:30), (●) DMPG:CL (60:40), and (●) CL in Buffer (20 mM Hepes, 500 mM NaCl and 1mM EDTA).

**Table 1 membranes-12-00534-t001:** Prediction of the secondary structure analysis of the peptide LL-37 in buffer and in different lipid systems.

Peptide/Lipid System	α-Helix prediction (%)
LL-37 in Hepes	32.0
LL-37 + DMPC:SM:DMPE	40.0
LL-37 + DMPG:CL	63.1

Prediction of secondary structure elements α-helix were performed using the methods supplied by the Confocheck^TM^ systems.

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
