# Peer review of "Infrared Spectroscopic Study of Multi-Component Lipid Systems: A Closer Approximation to Biological Membrane Fluidity"

_membranes, 2022, doi:10.3390/membranes12050534_

Round 1

Reviewer 1 Report

The manuscript is devoted to the study of the Fourier transform infrared spectroscopy to evaluate the thermal behavior of multicomponent supported lipid bilayers that mimic the membrane composition of microorganisms. I believe that this manuscript is interesting and should be publishable in this journal; however there are several scientific aspects of this manuscript that I feel the authors must first address.

  1. The introduction section does not provide complete information about the fluidity of the membrane and how it is organized. The organization of the introduction of the manuscript should be improved.
  2. Please, clarify the choice of LL-37 peptide in this work. Are your conclusions will be truth if you use another peptide?
  3. The organization of the manuscript should be improved. While some of the results are definitely interesting, the writing of the article needs to be substantially improved because it is not always clear what the authors are actually implying.
  4. Please, clarify the choice of lipids composition using the various methods. The surface charge affects the ability of LL-37 peptide and to fluidity of the membrane?
  5. Please, clarify the choice of the concentrations of LL-37 peptide using in the work. Why do not use the concentration range of the tested drugs? In this regard, it is necessary to discuss the detergent effects of compound on the properties of lipid membranes.
  6. Besides these fallacies, the discussion has to be improved and include more information about the membrane interaction mechanism of peptide and, consequently, effect on the fluidity of the membrane different composition
  7. Moreover, my great concern is related to the conclusions. They are too short and schematic. They should be modified. The most important findings of this work should be supported by experimental results and their biological significance should be clearly specified.

Reviewer 2 Report

The manuscript of Klaiss-Luna et al. describes the use of attenuated total reflection (ATR) infrared spectroscopy to measure the temperature of the gel-to-liquid crystalline phase transition of lipid membranes and uses this tool to probe the impact of lipid composition on the temperature dependence of this phase transition. The authors show that the presence of cholesterol, which is ubiquitously present in eukaryotic cell membranes, significantly broadens the phase transition. Overall, the manuscript is very well written and provides data of good quality. It touches an interesting topic (compositional complexity of membranes), which fits very well to the scope of Membranes. Nevertheless, the work can be improved with respect to the motivation of the investigations and the discussion of the data (with respect to what is known in the field) and not all conclusions appear to be well supported by the data presented (see below for details). Hence, I cannot yet recommend the manuscript for publication, but would like to see a revised version of the manuscript, which takes the following issues into account:

MAJOR ISSUES

  1. The abstract should be rewritten. Currently, it mainly contains information on the background of the study (lines 11-17), one sentence about the methodology (lines 17-20) and then one sentence about a result (lines 21-23), which is in my opinion not yet supported by the information given in the manuscript (see item 5 below). So it appears that the abstract in its current form does not reflect any of the results obtained by the authors with respect to the phase transitions studied in this work.
  2. Motivation (pages 1-2): The authors introduce fluidity probes and infrared spectroscopy as techniques that enable to measure the transition temperature at which the phase transition from gel-to-liquid crystalline phase occurs in bilayers. I miss an introduction of other techniques like differential scanning calorimetry (DSC) or dynamic light scattering (DLS), which are, in my opinion, much more often used to measure this transition temperature. In this sense, it remains unclear to me, which advantage is offered by ATR infrared spectroscopy in comparison to other available techniques (DSC and DLS are also label-free) and how well the presented results compare with the outcome of complementary techniques (see also item 3).
  3. The transition temperatures of single-component SLBs match well to complementary techniques. But what about the multicomponent systems? E. coli extracts have been well studied in the past using complementary techniques, which suggest the transition temperature to be rather 30° and not 45° (as in this study). The authors should also discuss this discrepancy and the relatively high transition temperatures of the model membrane observed in this study (for E. coli and P. aeruginosa as well as the cell models), which imply that these membranes are typically in a gel phase (as 45° are barely exceeded in nature) and thus are in conflict with studies showing the fluid nature of cell membranes.
  4. Page 4, line 138: The authors dry the lipids on an ATR crystal and then rehydrate them for the measurement. This for sure generates a lipid film at the interface, the structure of which can, however, be quite complex (single versus multiple bilayer etc.). Do the authors have any information on the properties/quality of the formed layer (thickness, density of defects, fluidity)? Even when using only a single lipid component, one may easily end up with a supported lipid bilayer (SLB) having a notable fraction of immobile lipids. I’m asking this, because the authors explicitly address the fluidity of SLBs (see e.g. the title of this work), while the transition measured by the authors may be just a local property of the lipids (conformation of the acyl chains) and actually not representative for the fluid state of the entire SLB.
  5. The authors conclude in the abstract that “The results showed that using multi-component lipid systems as model membranes for peptide-membrane interaction studies enables a better representation of the complexity of each cell membrane according to charge and fluidity properties.” What is the basis for this conclusion? The measurements involving the peptide LL-37 (page 8) focus on the structure of the peptide, but not on fluid aspects of the membrane (as claimed in the second part of this sentence). Furthermore, I would expect to see somewhere a comparison of single- with multi-component bilayers in order to draw such a conclusion, which is currently lacking (maybe I just missed it).
  6. Section 3.2: To me, it is totally unclear, why the authors included data on the secondary structure of a peptide here. So far, the entire manuscript focused on the impact of lipid composition on the phase transition and adding some secondary structure data taken at some (apparently arbitrarily chosen) lipid composition does not add knowledge in my opinion. Either the authors improve the motivation of these measurement or remove them from the manuscript.

MINOR ISSES

  1. Maybe a stupid question, but how is it possible to measure changed in the CH-peak of << 0.1 cm-1 with a detector offering a spectral resolution of 4 cm-1 (page 3, line 126). I guess that the peak are fitted by a certain model to enable this and hence, adding a representative series of IR spectra for one of the presented measurements could be helpful to the readers to understand the generation of the data.
  2. An interesting finding is the observation that the phase transition is less steep in presence of cholesterol (Figure 3). Was this also observed using complementary approaches (there is plenty of literature on lipid phase transitions)?
  3. Page 8, line 288: What do the authors mean with “cooperative” in this context? 

Round 2

Reviewer 1 Report

The authors have correctly addressed all the comments. The manuscript is recommend for publication.